Association between body mass index and blood pressure levels across socio-demographic groups and geographical settings: analysis of pooled data in Peru

http://orcid.org/0000-0002-6834-1376 Bernabe-Ortiz Antonio 1 2 antonio.bernabe@upch.pe
http://orcid.org/0000-0002-2090-1856 Carrillo-Larco Rodrigo M. 1 3
http://orcid.org/0000-0002-4738-5468 Miranda J. Jaime 1 4
1 CRONICAS Center of Excellence in Chronic Diseases, Universidad Peruana Cayetano Heredia , Lima , Peru
2 Universidad Cientifica del Sur , Lima , Peru
3 Department of Epidemiology and Biostatistics, School of Public Health, Imperial College London , London , United Kingdom
4 The George Institute for Global Health, University of New South Wales , Sydney , Australia
Palazón-Bru Antonio
Electronic publication date: 2021 Apr 21
Publication date: 2021
Volume: 9
Electronic Location ID: e11307
Received 2021 Jan 14; Accepted 2021 Mar 29
Copyright: © 2021 Bernabe-Ortiz et al.
Copyright year: 2021
Copyright holder: Bernabe-Ortiz et al.
License: This is an open access article distributed under the terms of the Creative Commons Attribution License, which permits unrestricted use, distribution, reproduction and adaptation in any medium and for any purpose provided that it is properly attributed. For attribution, the original author(s), title, publication source (PeerJ) and either DOI or URL of the article must be cited.
License URL: https://creativecommons.org/licenses/by/4.0/

Keywords: Blood pressure, Body mass index, Hypertension, Obesity

Funding: Wellcome Trust International Training Fellowship, London, UK 214185/Z/18/Z Rodrigo M. Carrillo-Larco is supported by a Wellcome Trust International Training Fellowship, London, UK (214185/Z/18/Z). The funders had no role in study design, data collection and analysis, decision to publish, or preparation of the manuscript.

==============================
Background

Understanding the relationship between BMI and blood pressure requires assessing whether this association is similar or differs across population groups. This study aimed to assess the association between body mass index (BMI) and blood pressure levels, and how these associations vary between socioeconomic groups and geographical settings.

Methods

Data from the National Demographic Health Survey of Peru from 2014 to 2019 was analyzed considering the complex survey design. The outcomes were levels of systolic (SBP) and diastolic blood pressure (DBP), and the exposure was BMI. Exposure and outcomes were fitted as continuous variables in a non-linear quadratic regression model. We explored effect modification by six socioeconomic and geographical variables (sex, age, education level, socioeconomic position, study area, and altitude), fitting an interaction term between each of these variables and BMI.

Results

Data from 159, 940 subjects, mean age 44.4 (SD: 17.1), 54.6% females, was analyzed. A third (34.0%) of individuals had ≥12 years of education, 24.7% were from rural areas, and 23.7% lived in areas located over 2,500 m above sea level. In the overall sample mean BMI was 27.1 (SD: 4.6) kg/m2, and mean SBP and DBP were 122.5 (SD: 17.2) and 72.3 (SD: 9.8) mmHg, respectively. In the multivariable models, greater BMI levels were associated with higher SBP (p-value < 0.001) and DBP (p-value < 0.001). There was strong evidence that sex, age, education level, and altitude were effect modifiers of the association between BMI and both SBP and DBP. In addition to these socio-demographic variables, socioeconomic position and study area were also effect modifiers of the association between BMI and DBP, but not SBP.

Conclusions

The association between BMI and levels of blood pressure is not uniform on a range of socio-demographic and geographical population groups. This characterization can inform the understanding of the epidemiology and rise of blood pressure in a diversity of low-resource settings.

Introduction

Globally, the number of adults living with raised blood pressure has increased from 594 million in 1975 to 1.13 billion in 2015 (NCD Risk Factor Collaboration (NCD-RisC), 2017a); and this condition is responsible for 9.4 million deaths worldwide (Lim et al., 2012), disproportionally affecting low- and middle-income countries. On the other hand, the alarming increase of body mass index (BMI) during the last decades (NCD Risk Factor Collaboration (NCD-RisC), 2017b), accompanied by the increase in the prevalence of overweight and obesity, will most likely pair the raise in blood pressure levels.

In Latin America, despite the high prevalence of cardiometabolic risk factors (NCD Risk Factor Collaboration (NCD-RisC)—Americas Working Group, 2020), their distribution is not homogenous between countries introducing additional particularities in shaping prevention and control strategies for raised blood pressure across settings. The increases in the levels of blood pressure and BMI are two ongoing global phenomena observed in recent decades, and understanding the patterning of its linkages is required to better counter them. This is of utmost importance as raises in BMI have been predominantly driven by increases in rural areas (NCD Risk Factor Collaboration (NCD-RisC), 2019).

BMI has been positively associated with both systolic (SBP) and diastolic (DBP) blood pressure (Drøyvold et al., 2005; Linderman et al., 2018). Although previous studies have described this association, many of these reports arise from high-income countries (Danon-Hersch et al., 2007; Gelber et al., 2007; Hubert et al., 1983), and from Asia and Africa (Linderman et al., 2018; Tesfaye et al., 2007), with limited representation from Latin American settings.

Understanding the relationship between BMI and blood pressure requires assessing whether this association is similar or differs across population groups. Peru, a middle-income country located in South America, is a country with pronounced socioeconomic disparities. These inequalities are paired with a unique geographical diversity with populations living across a range of low and high-altitude, and rural and urban areas. These sociodemographic factors can influence how the BMI is associated with blood pressure levels. Therefore, we aimed to assess the association between BMI and both SBP and DBP, and determine whether this association varies between different subgroups of the population.

Methods

Study design and population

Data from Peru’s National Demographic Health Survey (ENDES, Spanish acronym) from 2014 to 2019 was utilized in this analysis. The ENDES is a nationally representative population-based survey conducted annually. Since 2014, the ENDES has also included a health questionnaire and blood pressure measurements, on a yearly basis.

The ENDES sampling followed a bietapic approach. In rural areas, the primary sampling units were clusters of 500 to 2,000 individuals and the secondary sampling units were the households within each of these clusters. In urban areas, the sampling units were blocks or groups of blocks with more than 2,000 individuals and an average of 140 households; the secondary sampling units were also households as in rural settings. Details about the sampling process are described in detail elsewhere (Instituto Nacional de Estadistica e Informatica, 2015). For this analysis we included data from participants aged 20 years and above (as adolescence, i.e., up to 19 years, may affect body distribution and for instance BMI) with complete BMI and blood pressure measures.

Variables definition

Outcomes

The outcomes were both SBP and DBP. To guarantee that only plausible measurements were included in analyses, a cut point was used as in previous studies (NCD Risk Factor Collaboration (NCD-RisC), 2017a): SBP measurements should be ≥70 and ≤270 mmHg, whereas DBP measurements should be ≥50 and ≤150 mmHg. The no-plausible values were dropped from analyses.

Blood pressure was measured using a digital monitor (OMRON, model HEM-713), validated for adult populations (Takahashi, Yoshika & Yokoi, 2015). Two types of cuffs were used depending on the arm circumference of participants: standard arm (220 to 320 mm) and thicker arm (330 to 430 mm). Blood pressure measurements were assessed twice with the participant sitting and their right arm resting on a flat surface at the heart level. The first measurement was taken after a resting period of at least 5 min, and the second measurement was taken 2 min thereafter (Instituto Nacional de Estadistica e Informatica, 2018). The average of the two assessments was used in the analysis only if the difference between both measurements did not exceed 10 mmHg. Those exceeding that threshold were dropped from further analyses.

Exposure

BMI (kg/m2), as a continuous variable, was defined as weight (kg) divided by height squared (m2). Participants’ weight and height were measured following standard procedures (Instituto Nacional de Estadistica e Informatica, 2018), with the subject without shoes. To guarantee that only plausible BMI measurements were included in analyses, a cut point was used as in previous studies (Linderman et al., 2018): BMI ≥10 kg/m2 and <60 kg/m2.

Effect modifiers

A set of socio-demographic and geographical variables were also included as confounders and potential effect modifiers of the association of interest. We included sex (female vs. male); age (as continuous variable, but also categorized as <40, 40–59, and ≥60 years); education level (in years, <7, 7–11, and ≥12 years), and socioeconomic position (defined in tertiles based on a wealth index). This wealth index was calculated by the National Institute of Statistics and Informatics (INEI, Spanish acronym) based on assets and services that the participant reported having in the household (Rutstein & Johnson, 2004). As geographical variables we considered area (rural vs. urban) and altitude (≤500, 501–2,500, and ≥2,501 m above the sea level [m.a.s.l.]).

Other variables

For description of the participants characteristics in our study sample, we also provide information on hypertension, overweight and obesity. The ENDES collected self-reported information about whether the participant had a previous diagnosis of hypertension and if participants were receiving any anti-hypertensive medication. Therefore, hypertension was also defined as SBP ≥ 140 mmHg or DBP ≥ 90 mmHg, or having a previous diagnosis with hypertension (Chobanian et al., 2003). Standard BMI thresholds were used to estimate prevalence of overweight and obesity: BMI ≥ 25 and <30 kg/m2 for overweight, and BMI ≥ 30 kg/m2 for obesity, whereas BMI < 25 kg/m2 was classified as normal.

Statistical methods

Analyses were performed using STATA 16 for Windows (StataCorp, College Station, TX, US). Descriptive statistics and estimates were calculated considering the complex survey design using sample strata, primary sampling units and sampling weights, including the analysis of subpopulation groups (West, Berglund & Heeringa, 2008). We decided to pool data to maximize sample size while also decreasing statistical uncertainty; this process is consistent with large and global data pooling endeavors conducted before (e.g., IHME/GBD, NCD-RisC, among others).

Initially, description of continuous variables was conducted using mean and standard deviation (SD), whereas for categorical variables we used absolute and relative frequencies. Prevalence of hypertension, overweight, and obesity, and their respective 95% confidence intervals (95% CI), were also estimated. Comparisons were performed using Chi-squared test corrected for the survey design with the second-order correction of Rao and Scott (Rao & Scott, 1984) for categorical variables, whereas t-test was used for numeric variables.

To assess the association between BMI and blood pressure, a non-linear regression model (more details are provided in the Supporting Information file: Statistical approach) was built as suggested in literature (Kerry et al., 2005). An adjusted model was also created including coefficients for categorical variables (i.e., sex, education level, socioeconomic position, study area and altitude), whereas age was included as continuous variable as previous studies (NCD Risk Factor Collaboration (NCD-RisC)—Americas Working Group, 2020; NCD Risk Factor Collaboration (NCD-RisC), 2017a; NCD Risk Factor Collaboration (NCD-RisC), 2017b).

To explore effect modification, i.e., to evaluate whether the association between BMI and levels of blood pressure vary by specific population groups, i.e., by socio-demographic and geographical variables, an interaction term between each socio-demographic variable and BMI (and its quadratic term) was included (see Supporting Information file: Statistical approach). Effect modification was assessed using the testparm, a post-estimation command in STATA and a p-value < 0.05 was considered significant. When conducting multivariable model stratified by subgroups, we excluded the variable defining that subgroup from the list of potential confounders. Collinearity was evaluated using the tolerance test in STATA. Marginal means of blood pressure levels for specified fixed values of socio-demographic variables were calculated from each fitted model and presented as figures (and also tables) to verify variation of the association of interest (Graubard & Korn, 1999; Williams, 2012). Values of BMI higher than 50 kg/m2 were truncated to avoid the impact of few values on results (<0.1% of subjects had values of BMI > 50 kg/m2).

Finally, sub-analyses were also conducted to assess the association of interest (overall and by sub-population group) by excluding individuals who reported taking anti-hypertensive medication as this could bias results.

Ethics

The data used in our study is anonymous and freely available to the general public and does not reveal any personal identifiable information, and consequently this study does not represent an ethical risk for participants. The INEI, a Peruvian government organization, is responsible for the annual collection of ENDES data. This institution requested the consent of participants before the survey.

Results

Characteristics of the study population

Data from a total of 203,880 individuals was available for the analysis utilizing ENDES annual evaluations over the study period. Of them, information from 22,646 (11.1%) aged <20 years; 2,734 (1.3%) pregnant women; 15,519 (7.6%) without blood pressure; 3,007 (1.5%) without BMI results, and 34 (<0.1%) with BMI ≥ 60 kg/m2 were excluded from our analysis. Thus, a total of 159,940 subjects, mean age 44.4 (SD: 17.1), 54.6% females, were analyzed. Of note, only a third (34.0%) of individuals had ≥12 years of education, 24.7% were from rural areas, and 23.7% lived in areas located over 2,500 m.a.s.l.

Mean BMI was 27.1 (SD: 4.6) kg/m2 in the overall sample. The prevalence of overweight was 41.6% (95% CI [41.2–42.0%]), and the prevalence of obesity was 23.5% (95% CI [23.1–23.9%]). Similarly, mean SBP and DBP were 122.5 (SD: 17.2) and 72.3 (SD: 9.8) mmHg, respectively, whilst the prevalence of hypertension was 21.2% (20.8–21.6%).

When comparing socio-demographic characteristics by sex, males had higher number of years of education and greater blood pressure levels and hypertension rates compared to females. Conversely, females had higher body mass index and greater obesity rates (Table 1).

Table 1 Description of the study population by sex comparison considering complex sampling design.

	Females	Males	p-value	
	(n = 89,376)	(n = 70,538)	
Age				
Mean (SD)	44.3 (16.9)	44.4 (17.2)	0.51	
Age (categories)				
<40 years	51,232 (46.5%)	36,727 (45.8%)	0.31	
40–59 years	24,553 (33.1%)	22,188 (33.5%)		
≥60 years	13,591 (20.4%)	11,623 (20.7%)		
Education level				
<7 years	27,034 (30.4%)	18,093 (23.4%)	<0.001	
7–11 years	32,046 (36.5%)	30,114 (41.6%)		
≥12 years	24,479 (33.1%)	20,970 (35.0%)		
Socioeconomic position (assets)				
Low	29,107 (23.6%)	24,070 (24.6%)	0.002	
Middle	30,201 (30.1%)	23,557 (30.3%)		
High	30,068 (46.3%)	22,911 (45.1%)		
Study area				
Rural	30,142 (24.1%)	25,632 (25.4%)	<0.001	
Urban	59,234 (75.9%)	44,906 (74.6%)		
Altitude				
≤500 m.a.s.l.	43,316 (59.9%)	34,757 (61.6%)	<0.001	
501–2500 m.a.s.l.	17,850 (15.4%)	14,805 (16.0%)		
≥2501 m.a.s.l	28,210 (24.7%)	20,976 (22.4%)		
Body mass index				
Mean (SD)	27.5 (5.0)	26.6 (4.4)	<0.001	
Body mass index (categories)				
Normal (<25 kg/m2)	30,048 (32.4%)	29,208 (38.0%)	<0.001	
Overweight (25 to <30 kg/m2)	36,006 (40.4%)	29,276 (43.0%)		
Obese (≥30 kg/m2)	23,322 (27.2%)	12,054 (19.0%)		
Blood pressure levels				
Systolic blood pressure, mean (SD)	118.4 (16.7)	127.4 (15.3)	<0.001	
Diastolic blood pressure, mean (SD)	70.3 (9.3)	74.7 (9.9)	<0.001	
Hypertension				
Yes	13,690 (19.5%)	13,660 (23.2%)	<0.001	
Note:

SD, Standard deviation; m.a.s.l. = meters above the sea level.

Association between body mass index and blood pressure levels

Greater BMI levels were associated with increasing SBP (p-value <0.001) and DBP (p-value <0.001) values in multivariable models (Figs. 1A–1B).

Figure 1 Body mass index and (A) systolic and (B) diastolic blood pressure.

The increase of SBP per unit BMI varied from 0.05 to 1.8 mmHg/(kg/m2), whereas the increase of DBP ranged from 0.09 to 1.13 mmHg/(kg/m2). Estimates for specific BMI cut points are shown in Table 2 and Table 3 (by sex), where the highest values of both, SBP and DBP, were reached at BMI of 45 kg/m2.

Table 2 Estimation of blood pressure levels according to body mass index (BMI): in the total sample and excluding those with anti-hypertensive medication.

BMI (in kg/m2)	In the overall sample	Only those without antihypertensive medication	
	Systolic blood pressure (mm Hg)	
10	98.5 (97.2–99.9)	98.5 (97.2–99.8)	
15	107.0 (106.3–107.7)	106.8 (106.1–107.5)	
20	114.2 (113.9–114.5)	113.7 (113.4–114.0)	
25	120.0 (119.8–120.2)	119.4 (119.2–119.6)	
30	124.5 (124.3–124.7)	123.8 (123.6–124.0)	
35	127.7 (127.4–128.0)	126.9 (126.6–127.1)	
40	129.6 (128.9–130.2)	128.6 (128.0–129.2)	
45	130.1 (129.0–131.2)	129.1 (128.0–130.2)	
50	129.3 (127.4–131.2)	128.3 (126.5–130.1)	
	Diastolic blood pressure (mm Hg)	
10	56.2 (56.4–58.0)	57.1 (56.3–58.0)	
15	62.5 (62.1–63.0)	62.4 (62.0–62.9)	
20	67.1 (66.9–67.3)	67.0 (66.8–67.1)	
25	70.9 (70.8–71.0)	70.7 (70.6–70.8)	
30	73.9 (73.8–74.0)	73.7 (73.6–73.8)	
35	76.1 (75.9–76.2)	75.9 (75.7–76.1)	
40	77.5 (77.1–77.8)	77.3 (76.9–77.7)	
45	78.1 (77.4–78.8)	78.0 (77.2–78.7)	
50	77.9 (76.7–79.0)	77.8 (76.6–79.0)	
Note:

Marginal estimates were built on adjusted models.

Table 3 Estimation of blood pressure levels according to body mass index (BMI) by sex.

BMI
(in kg/m2)	Females	Males	
In the overall sample	Only those without antihypertensive medication	In the overall sample	Only those without antihypertensive medication	
	Systolic blood pressure (mm Hg)	
10	97.0 (95.3–98.6)	97.4 (95.8–99.0)	100.2 (98.0–102.4)	99.6 (97.5–101.7)	
15	104.2 (103.3–105.1)	104.0 (103.2–104.9)	110.2 (109.1–111.4)	109.8 (108.8–110.9)	
20	110.4 (110.0–110.8)	109.8 (109.4–110.2)	118.6 (118.1–119.0)	118.3 (117.9–118.8)	
25	115.5 (115.3–115.8)	114.6 (114.4–114.8)	125.2 (125.0–125.5)	125.0 (124.8–125.2)	
30	119.7 (119.4–119.9)	118.6 (118.3–118.8)	130.2 (129.9–130.5)	129.9 (129.6–130.2)	
35	122.7 (122.4–123.1)	121.6 (121.2–121.9)	133.6 (133.1–134.1)	133.1 (132.6–133.6)	
40	124.8 (124.0–125.5)	123.6 (122.9–124.4)	135.2 (134.1–136.3)	134.4 (133.4–135.5)	
45	125.8 (124.4–127.1)	124.8 (123.5–126.2)	135.2 (133.2–137.2)	134.1 (132.1–136.0)	
50	125.8 (123.5–128.0)	125.1 (122.9–127.3)	133.5 (130.2–136.8)	131.9 (128.8–135.1)	
	Diastolic blood pressure (mm Hg)	
10	56.8 (55.7–57.8)	57.0 (55.9–58.0)	58.1 (56.7–59.5)	57.7 (56.3–59.0)	
15	61.5 (60.9–62.0)	61.5 (60.9–62.0)	64.0 (63.3–64.7)	63.7 (63.0–64.4)	
20	65.5 (65.2–65.7)	65.3 (65.0–65.6)	69.1 (68.8–69.4)	68.9 (68.6–69.2)	
25	68.8 (68.7–68.9)	68.5 (68.4–68.7)	73.3 (73.2–73.5)	73.2 (73.1–73.4)	
30	71.4 (71.3–71.6)	71.1 (71.0–71.3)	76.8 (76.6–77.0)	76.7 (76.5–76.9)	
35	73.4 (73.2–73.6)	73.1 (72.9–73.3)	79.4 (79.1–79.8)	79.3 (78.9–79.6)	
40	74.7 (74.2–75.1)	74.5 (74.0–74.9)	81.2 (80.5–82.0)	81.0 (80.2–81.8)	
45	75.3 (74.5–76.1)	75.2 (74.4–76.1)	82.2 (80.8–83.7)	81.9 (80.4–83.3)	
50	75.2 (73.8–76.5)	75.3 (73.9–76.7)	82.4 (80.1–84.7)	81.8 (79.5–84.1)	
Note:

Marginal estimates were built on adjusted models.

The association between BMI and SBP differed by different socio-demographic variables(Figs. 2A–2F), it was stronger among males (p-value for interaction term <0.001), among older individuals (p-value for interaction term <0.001), those less educated (p-value for interaction term = 0.001), and among those living under 500 meters above sea level (p-value for interaction term = 0.005). On the other hand, all the socio-demographic and geographical variables studied were effect modifiers of the association between BMI and DBP (Figs. 3A–3F): it was stronger among males (p-value for interaction terms <0.001), among older individuals (p-value for interaction term <0.001), better educated (p-value for interaction terms <0.001), those with low socioeconomic position (p-value for interaction terms <0.001), those in urban areas (p-value for interaction terms <0.001), and among those living at high altitude (p-value for interaction terms <0.001). Coefficients and 95% CI are shown in the Supporting Information file: Tables S1 to S7. All the tolerance values for collinearity were <5.

Figure 2 Association between body mass index and systolic blood pressure according to socio-demographic variables.

(A) By sex, (B) by age, (C) by education level, (D) by socioeconomic position, (E) by area, and (F) by altitude.

Figure 3 Association between body mass index and diastolic blood pressure according to socio-demographic variables.

(A) By sex, (B) by age, (C) by education level, (D) by socioeconomic position, (E) by area, and (F) by altitude.

When the analyses were conducted excluding subjects taking anti-hypertensive medication, results and figures were similar to those shown as main results as only about a third (30.4%) of participants with hypertension reported taking antihypertensive pharmacological medication (Details in the Supporting Information file: Tables S8 to S12).

Discussion

Using nationally representative data from Peru, accrued over five years in approximately 160,000 subjects, we advance the evidence reported on the association between BMI and blood pressure by exploring the effect of relevant socio-economic health determinants, namely sex, age, education level, socioeconomic position, study area, and altitude, much needed for adequate resource allocation. Whilst the effect modification in the relationship between BMI and DBP was present in all of these subgroups, socioeconomic position and study area were not effect modifiers in the association between BMI and SBP.

The global rise of levels of BMI and blood pressure calls for paying attention at particular groups; hence, our results disaggregated by socio-demographic groups show clear patterns of association between SBP and DBP when BMI increases. These results, for countries in transition, can inform tailored prevention and control strategies to counter the rise in levels of blood pressure and BMI, focusing on specific groups to yield higher prevention gains and further population-wide benefits. It is notorious the role of sex, but the peculiarities of area, including living in high altitude areas, as well as education, may be also important. Interventions or treatment allocation to improve BMI and blood pressure may not need different approaches by different socio-demographic variables, but could benefit of gender-based tailoring to maximize its impact.

Globally, mean blood pressure levels are higher in African countries, and Peru ranks among the countries with lowest mean blood pressure in the world (NCD Risk Factor Collaboration (NCD-RisC)—Americas Working Group, 2020; NCD Risk Factor Collaboration (NCD-RisC), 2017a). Much of the literature on the relationship between BMI and blood pressure arises from Asia and some parts of Africa (Danon-Hersch et al., 2007; Tesfaye et al., 2007). Importantly, a nationwide study of 1.7 million Chinese adults explored similar relationships between BMI and blood pressure across a diversity of groups (Linderman et al., 2018), with some notorious observations contrasting our findings. Compared to our Peruvian sample, Chinese individuals had approximately 2.5 kg/m2 units lower than Peruvians (24.7 vs. 27.1 kg/m2), yet their SBP was 14 mmHg higher (136.5 vs. 122.4 mmHg), and DBP was 9 mmHg higher (81.1 vs. 72.2 mmHg). While both studies confirm a relationship between BMI and blood pressure across a range of groups, the trajectories of SBP values in Peruvians range between 100 and 140 mmHg in all the groups studied (Fig. 2), whereas Chinese SBP values reach up to 150 mmHg and 160 mmHg in some cases (Linderman et al., 2018). In Latin America, one previous study from Brazil, focused on a single state, analyzed the association between two anthropometric indexes, waist circumference and BMI, and hypertension and found both indexes associated with hypertension, but showed a low sensitivity as predictors (Peixoto Mdo et al., 2006).

Our observations, benefiting from nationwide representative data, are useful not only in describing the patterns of the relationships between BMI and blood pressure, but also in adequately informing policies. Such tailored policies are needed as premature non-avertable mortality from non-communicable diseases decreased slightly or stagnated in most regions, and rose in the region of the Americas (Martinez et al., 2020). Therefore, characterizing the patterns of the BMI-blood pressure relationship are crucial in the Latin American region, particularly given the high level of heterogeneity in the transition of major cardiovascular risk factors in the region (NCD Risk Factor Collaboration (NCD-RisC)—Americas Working Group, 2020).

Peru is a middle-income country located in South America, with a fragile and fragmented health care system, with poor response to the challenges of chronic conditions, and for instance, may not be prepared to tackle the burden of increasing BMI and blood pressure levels (Loret de Mola et al., 2014). A recent manuscript demonstrated an increasing age-standardized prevalence of hypertension in the last years in Peru, whereas the rates of disease awareness and controlled hypertension have declined (Villarreal-Zegarra, Carrillo-Larco & Bernabe-Ortiz, 2020). Our results demonstrated slightly change in blood pressure estimates in the overall sample and when excluding those with anti-hypertensive medication pointed out the low rates of appropriate treatment and control (Zavala-Loayza et al., 2016). Thus, a thoroughly planned and balanced policy would be needed to provide care to those who most needed, including appropriate hypertension diagnosis and treatment.

This study benefits from data derived from a nationally representative survey of adult population, with objective measurements, and a robust modeling approach that goes beyond linear associations. Our study expands to disentangle relationships across a diversity of socioeconomic and geographical population groups, much needed to characterize the epidemiology of BMI and blood pressure, and to inform policies. Despite this, we are aware of certain limitations including the cross-sectional nature of the data, together with potential for reverse causality. Given that both, obesity and high blood pressure are established global public health priorities, both require urgent action to prevent their associated complications. Also, the graphical nature of the BMI-blood pressure trajectories should be interpreted with caution as comparisons of levels of blood pressure across BMI groups are not necessarily correct, but rather the result of calculations of blood pressure values given a value of BMI. Whilst we cannot ascertain that a given group has a number of blood pressure units, higher or lower, than other group, the trajectories are useful in identifying potential groups to target action, e.g., males compared to females. In addition, across the years, the sampling frame of the ENDES survey has changed; for example, the sampling frame of the last two years is not the same as in the previous years. However, where relevant, we used the year-specific sampling weights, stratum and sampling units. Other variables (e.g., dietary consumption, physical activity, and comorbidities) were not included in the models as they were not available.

Conclusions

In conclusion, greater BMI is associated with higher blood pressure levels in Peru, but this association is not uniform across a range of socio-demographic and geographical population groups. These results may contribute to prevention and control strategies to counter the rise in levels of blood pressure and BMI, focusing on specific groups to yield higher prevention gains and further population-wide benefits.

Supplemental Information

Supplemental Information 1 Supporting information.

Click here for additional data file.

Additional Information and Declarations

Competing Interests

Author Contributions

Data Availability

The authors declare that they have no competing interests.

Antonio Bernabe-Ortiz conceived and designed the experiments, performed the experiments, analyzed the data, prepared figures and/or tables, authored or reviewed drafts of the paper, and approved the final draft.

Rodrigo M. Carrillo-Larco conceived and designed the experiments, performed the experiments, analyzed the data, prepared figures and/or tables, authored or reviewed drafts of the paper, and approved the final draft.

J. Jaime Miranda performed the experiments, prepared figures and/or tables, authored or reviewed drafts of the paper, and approved the final draft.

The following information was supplied regarding data availability:

The data used in our study is anonymous and freely available to the general public and does not reveal any personal identifiable information, and consequently this study does not represent an ethical risk for participants. The INEI, a Peruvian government organization, is responsible for the annual collection of ENDES data. This institution requested the consent of participants before the survey. Data is available at http://iinei.inei.gob.pe/microdatos/.

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
