# Peer review of "Association between body mass index and blood pressure levels across socio-demographic groups and geographical settings: analysis of pooled data in Peru"

_PeerJ, doi:10.7717/peerj.11307_

## Round 0.1 · original submission · Major Revisions

Two reviewers have analyzed your manuscript and they have found scientific merit in your work. However, there are some issues which you must address in a revised version of the text.

·

Basic reporting

The study Association between body mass index and blood pressure levels across socio-demographic groups and geographical settings: analysis of pooled data in Peru seems interesting to me. The aim of this study is to assess the association between body mass index (BMI) and blood pressure levels, and how these associations vary between socioeconomic groups and geographical settings.

Experimental design

No comments

Validity of the findings

Mayor concern

It is not clear how the statistical models were constructed.

It would be important to know the models with their beta values and their statistical significance values.

In the supplementary document and specifically in model three, when the effect modification related with sex was evaluated, I wonder why the variable sex alone and its corresponding Beta value were not included as recommended. What statistical criteria were used to decide that there is an effect modification?

The authors did not mention why did not evaluate the presence of collinearity in the different models. Nor do authors mentioned anything in relation to the assumptions of the regression used.

I think that the section on limitations is needed, for example, would have been very enriching to include other types of variables such as dietary consumption, physical activity and some comorbidities as adjustment variables, since they are variables that are always measured almost as a rule in surveys.

Minor concern:
Check the numbers, since they do not match: Excluded 21,294, total evaluated, total analyzed 159,940

Regarding Tables 2 and 3, I suggest clarifying whether they are unadjusted estimates.

I think that in the discussion section it is necessary to include information on the possible mechanisms of how BMI causes the systolic and diastolic pressure levels to rise, supported in the literature.

Additional comments

The study Association between body mass index and blood pressure levels across socio-demographic groups and geographical settings: analysis of pooled data in Peru seems interesting to me. The aim of this study is to assess the association between body mass index (BMI) and blood pressure levels, and how these associations vary between socioeconomic groups and geographical settings.

Mayor concern

It is not clear how the statistical models were constructed.

It would be important to know the models with their beta values and their statistical significance values.

In the supplementary document and specifically in model three, when the effect modification related with sex was evaluated, I wonder why the variable sex alone and its corresponding Beta value were not included as recommended. What statistical criteria were used to decide that there is an effect modification?

The authors did not mention why did not evaluate the presence of collinearity in the different models. Nor do authors mentioned anything in relation to the assumptions of the regression used.

I think that the section on limitations is needed, for example, would have been very enriching to include other types of variables such as dietary consumption, physical activity and some comorbidities as adjustment variables, since they are variables that are always measured almost as a rule in surveys.

Minor concern:
Check the numbers, since they do not match: Excluded 21,294, total evaluated, total analyzed 159,940

Regarding Tables 2 and 3, I suggest clarifying whether they are unadjusted estimates.

I think that in the discussion section it is necessary to include information on the possible mechanisms of how BMI causes the systolic and diastolic pressure levels to rise, supported in the literature.

Reviewer 2 ·

Basic reporting

See below

Experimental design

See below

Validity of the findings

See below

Additional comments

The authors aimed to assess the association between body mass index and blood pressure levels across socio-demographic groups and geographical settings, by using data from different Peruvian Demographic Health Surveys. I would like to congratulate the authors for such an interesting topic and its novel approach. However, I believe that some aspects should be addressed in order to improve the final version.

Major comments:
1. I understand that the ENDES have different sampling frames for a group of years (eg. the sampling frame for 2018-2020 is different from other years). Polling data from years with different sampling frames could lead to incorrect estimates due to problems in weighting (at least this was considered on the first steps of the analysis. If so, please note the process for polling the data that you have performed)
2. How did you perform the t-test for svy? In addition, it is important to mention that the usual version of the Chi2 (pearson) is not adequate for svy analysis. The authors might have used the Rao-Scott chi-square test.

Minor comments:
Introduction
Lines 68-72: This sentence is too long. Please split it on -at least- two.

Methods:
Line 88: Please explain why you included adults aged 20 and over (and not 15 and over)
Lines 112-113: Please explain why you decided to categorize the education level in years instead of primary, secondary, university...

---

## Round 0.2 · accepted · Accept

All the reviewers' questions have been correctly assessed in this revised version of the text.

·

Basic reporting

No comments. My mayor concern were taken into account.

Experimental design

No comments

Validity of the findings

No comments

Additional comments

No comments

Reviewer 2 ·

Basic reporting

See below

Experimental design

See below

Validity of the findings

See below

Additional comments

The authors have responded to all comments and suggestions. I consider that the manuscript is now suitable for publication.